# Perception of Size and Mass Relationships of Moving and Stationary Object in Collision Events in 10-to-11-Month-Old Infants

**DOI:** 10.3390/bs13010056

**Published:** 2023-01-06

**Authors:** Nilihan E. M. Sanal-Hayes, Lawrence D. Hayes, Peter Walker, Jacqueline L. Mair, James Gavin Bremner

**Affiliations:** 1Department of Psychology, Lancaster University, Lancaster LA1 4YW, UK; 2Sport and Physical Activity Research Institute School of Health and Life Sciences, University of the West of Scotland, Glasgow G72 0LH, UK; 3Future Health Technologies, Singapore-ETH Centre, Campus for Research Excellence and Technological Enterprise, Singapore 138602, Singapore

**Keywords:** causal events, collision events, looking time, mass cues, object size of moving object, object size of stationary object, physical knowledge, violation of expectation

## Abstract

Around 5.5–6.5 months of age, infants first attend to object size and perceive its mass cues in simple collision events. Infants attend to the size of the moving object and expect a greater displacement following a collision with a large object and stationary object, and lesser displacement following a collision with a small object and stationary object. It has been proposed that infants of 6-to-7 months of age can differentiate between sizes of moving objects but do not perceive the size and mass relationships in simple collision events. The present two investigations aimed to investigate whether infants 10-to-11 months of age (N = 16) could perceive this relationship (experiment 1) and the reverse of this relationship (experiment 2) utilising the looking time paradigm. The reverse of this relationship entailed the circumstances in which the moving object size was kept constant, but the stationary object size varied (small or large). Results from these experiments revealed that infants did not differ in their looking times for size congruent and size incongruent distances in both conditions. Infants did not look longer at the incongruent test events that violated expectation. For that reason, we conclude infants of 10-to-11 months of age were unable to perceive size and mass associations in collision events in either direction (moving object or stationary object size).

## 1. Introduction

The visual world is formed by objects of various physical properties such as size, shape, colour, surface material and other dimensional measurements that account for density and volume [1,2,3]. All these aforementioned object properties cue for object weight and mass in adults [4,5,6,7,8,9,10,11,12,13]. Size cues for mass, with the principle in mind that larger objects are usually perceived to be heavier in mass than smaller objects [14]. Object colour in turn, has an effect as luminance cues mass; darker objects are perceived to be heavier in mass than lighter objects [8]. Surface material is influential in that denser materials (e.g., steel and marble) are generally perceived to be of greater mass than less dense materials (e.g., wood). Perceived density affects mass judgement as objects with higher densities have greater mass per unit volume and thus are heavier than objects of the same volume with lower densities [5,10,11]. These object properties cue for mass when motor actions are involved due to the application of gravity, and for object mass when the amount of matter is concerned for example in visual perception of objects [15,16]. In a simple collision event designed to examine physical reasoning about causal events involving a moving object A and a stationary object B, the causal behaviour and interaction between object A and object B is attributed to knowledge of object properties [17,18,19,20,21,22].

Humans are born with functioning oculomotor (eye movement) processes [23]. Already when newborn, they perceive differences between shapes (e.g., Triangles, circles, crosses and squares) and various line orientations [24,25]. For example, newborn infants demonstrate preference for novel shapes (demonstrated by longer looking time) when compared to familiar shapes seen previously. Longer looking time for novel shapes compared to familiar shapes suggests infants can distinguish between shapes they have seen and not previously seen [24]. Similarly, neonates show preference for novel line orientations, indicated by their longer looking time for these stimuli compared to looking time for a line orientation seen previously [25]. Besides these accomplishments, neonates are also successful in perceiving shape or size of an object as constant despite differences in the angle of viewing [26,27]. For example, after watching an event in which the object is of constant size or shape, but the distance differs across trials, newborn infants demonstrate longer looking time at stimuli of different size or shape compared to objects of same size or shape at a different distance. This indicates that infants can detect changes in size or shape unaffected by the change of distance across stimuli [26,27]. Yet, there is some uncertainty concerning what age humans start to perceive object weight and object mass of various object physical properties in collision events. 

Some authors posit that infants perceive a collision between a moving and stationary object based on force of the moving object [28]. Furthermore, this force expectation of the moving object is claimed to help infants in their perception of smaller and larger moving objects involved in physical events [28,29]. Particularly, that larger objects in relation to smaller objects are more likely to exert greater force and result in greater displacements of a stationary object [28,30,31]. For example, infants of 5.5–6.5 month of age expect a larger moving object to propel a stationary object further than a smaller moving object [20]. This force expectation of the moving object has also been demonstrated to help infants detect the direction of collision events [29,32,33]. For example, infants are able to detect the reversal of the collision events (i.e., when the stationary object causes the moving object to move in the opposite direction by 6 months of age [29,32,33]). This detection suggests that infants predict the moving object to set the stationary object into motion through the force expectation of the moving object [29]. Yet, there are a paucity of data concerning infants’ perception of force in collision events. One particular unknown concerns how infants respond to events in which same force is exerted by the moving object to stationary objects of varying sizes. Infants’ perception of force of smaller and larger objects in collision events has previously only been examined in the context of the size of the moving object [20]. Manipulation of the size of the stationary object has not been considered in the infant literature to our knowledge. The scope of this paper was therefore to investigate infants’ perception of size of the stationary object in collision events on the outcome of said collision. This relationship in collision events is also easier to perceive due to encoding differences. For example, the stationary object is present during the entire collision event and the size of the cube (stationary object) can be contrasted with the other cubes below the ramp. Moreover, the moving object that is off view at certain times has a constant size throughout all events. However, the experiment that investigated infants’ perception of size of the moving object differ in that the moving object is not present at the start of the test event, but later presented by a hand that puts the ball on the ramp before it rolls down. This might not give infants enough time to encode and distinguish between the sizes of the moving object and the two other objects below the ramp. Moreover, it is a more simplified version than the collision events that manipulate the size of the moving object, because it is more complex to relate size of one moving object to distance propelled of another object (moving object size) as opposed to relate size of one stationary object to distance propelled of this same object (stationary object size).

Size is not independent on mass; size cues mass with the assumption that larger objects are often heavier in mass than smaller objects [14]. For that reason, it makes sense that heavier objects exert greater force and move other objects further in a collision regardless of direction [14,20]. However, this assumption is not true in cases when the moving object is constant in size and the stationary object varies in size. In these cases, the force exerted by the moving object is same. Infants should therefore infer the outcome of the collision event based on the size differences of stationary objects. In other words, infants should expect a stationary object of small size to be propelled further than a stationary object of large size by a consistent-size moving object. In Kotovsky and Baillargeon’s [20] experiment, infants demonstrated the ability to infer size of moving objects on the propelled distance of the stationary object. Hohenberger et al. [34] was successful in replicating this study of Kotovsky and Baillargeon [20] with 10-to-11-month-old infants using computer-generated collision events. For that reason, it made sense to test whether infants could transfer this ability to infer size of a moving object to infer size of a stationary object and expect certain outcomes of collision events based on the size of the moving and the stationary object. In the experiments by Sanal-Hayes et al. [35], they expanded the Kotovsky and Baillargeon study [20] and examined how infants perceived a large and small object displacing a stationary object to two distances (short and far). In their study, they highlighted that infants of 6-to-7 months of age can differentiate between the sizes of moving objects, but do not perceive the size and mass relationships in simple collision events. Infants of 6-to-7 months of age displayed longer looking times at large moving objects compared to small moving objects, but did not differ in their looking times for incongruent and congruent test events. The differences between these study outcomes can be attributed to methodological differences between the studies [19,20,35,36]. For example, the collision events differed between the two studies, one utilized real-life collision events, whereas the other utilized computer-generated collision events. The results might then differ across these studies due to various reasons such as 6-to-7-month-old infants’ failure to perceive the 3D computer-generated objects as violating expectations compared to 3D real-life objects or the lack of time permitted to encode the events on the display [37]. Infants of 6-to-7 months of age might still attend and be familiar with object size despite their failure to perceive the impossible events in the computer-generated collision events [35,36]. This might explain why the looking time for the impossible events do sometimes demonstrate a familiarity to the event [37]. Furthermore, the complexity of the collision events also differed, one examined two sizes on one travelled distance of stationary object, whereas another examined two sizes on two travelled distances of stationary object. This might again be too complex for an infant audience of 6-to-7 months of age and contain too much information to process.

In the present study, we used the experimental setup by Sanal-Hayes et al. [35,36]. We first investigated whether 10-to-11-month-old infants perceived the relationship of size of moving object (small or large) and displacement of a consistently sized stationary object to short or far distance in collision events (experiment 1). Next, we investigated whether 10-to-11-month-old infants perceived the relationship of size of stationary object (small or large) and displacement of a stationary object following a collision with a consistently sized moving object to short or far distance in collision events (experiment 2). In the second experiment, infants were habituated to an event in which a mid-size billiard ball rolled down a ramp and propelled a mid-size cube to midpoint of the screen. Next, infants saw a mid-size billiard ball roll down a ramp and propel either a large or small-size cube to before midpoint or endpoint of the screen. We hypothesised a priori that infants in both experiments would look longer at events that violated expectations of the size-congruent distances. In this context, infants should have looked longer when the large moving object displaced a size-constant stationary object to a short distance (experiment 1) or a large cube was displaced to a longer distance after a collision with a size-constant moving object (experiment 2), and when the small moving object displaced a size-constant stationary object to a far distance (experiment 1) or a small cube was displaced to a short distance after a collision with a size-constant moving object (experiment 2). The looking time paradigm dictates infants look longer at events which violate expectations. 

## 2. Experiment 1 Methods

### 2.1. Participants

A total of 32 infant participants took part in the study, but due to equipment failure (N = 1), failure to habituate (N = 12) and successful habituation but failure to watch test events (N = 3), the final sample consisted of 16 participants. Infants that failed to habituate did not display a decreased responsiveness to repeated stimuli (i.e., their looking time did not decrease when viewing the same event repeatedly). Furthermore, infants that successfully habituated but failed to watch the test events displayed a looking time duration that was shorter than the duration of the collision event taking place, this meant the infant did not see the full collision event. The 16 participants were between ages 304 days to 335 days (*M* = 321, *SD* = 11). All participants were recruited from the database at Lancaster University Babylab. Participants were healthy, full-term infants and received a book for their participation alongside being reimbursed for travel costs. Of these 16 infants, eight were female (age in days: *M* = 320, *SD* = 12) and eight were male (age in days: *M* = 323, *SD* = 10).

### 2.2. Materials and Apparatus

Computer-generated collision events were created using Animate C.C (2016), Adobe Systems. participants watched dynamic collision events on a screen. The backdrop consisted of an image of a wooden table (W = 20.89 cm, H = 5.07 cm), background of three houses (W = 13.81 cm, H = 5.50 cm), a ramp (W = 3.92 cm, H = 2.51 cm), a cube (W = 2.51 cm, H = 2.51 cm), a hand (W = 2.51 cm, H = 2.01 cm), a small ball (H = 1.06 cm, W = 1.06 cm), medium ball (H = 1.59 cm, W = 1.59 cm), and large ball (W= 2.38 cm, H= 2.38 cm). 

Infants watched the habituation event (see Figure 1 and Figure 2.) before the test events, in this event the cube propelled by a mid-size ball to the midpoint position (habituation distance). In the test events, saw the cube propelled by a large or small ball to a position either before the midpoint (shorter distance) or at the endpoint (longer distance) of screen. In the habituation event, the cube was propelled by a ball of physical properties cuing mid-mass (mid-size ball) to one distance (midpoint). In the test events, the cube was either propelled by a ball of physical properties cuing greater mass (large ball) or lesser mass (small ball). Test events showed a large ball or small ball propel the grey cube to a size-appropriate distance (congruent) and a size-inappropriate distance (incongruent). In the congruent outcomes the small ball propelled the cube to before the midpoint, and the large ball propelled the cube to the endpoint of the screen. In the incongruent outcomes the small ball propelled the cube to endpoint (long distance) and the large ball propelled the cube to before midpoint (short distance). As the 2D size of the balls were different (i.e., small, mid-sized, large), the *inferred* volumes of balls were therefore different.

Participants were shown test event (see Figure 2) scenes in which a hand was presented but the ball was hidden (for 1 s). Subsequently, the hand was hidden and then visible again holding the ball (for 1 s). The hand placed the ball on the ramp, pressed it down, and after 1 s, the hand was lifted. The ball rolled down the ramp (for 1 s) and propelled the cube in front of the first house to either the end of the first house or midpoint or to the last house (for 1–2 s). These events continued 1 s after movement ended to allow participants time to perceive the event in its entirety. In total, events in which the cube propelled to the end of the first house or midpoint lasted 6 s (240 frames, 48 frames/s), and events in which the cube propelled to the last house lasted 7 s (288 frames, 48 frames/s). The cube travelled 1.5 cm/s from the start of the first house to the end of the first house (shorter condition) or to midpoint (midpoint condition). The cube travelled 1.17 cm/s from the start of the first house to the middle of the third house (longer condition). 

The habituation event was showed in a loop of 9 trials until test events were shown. Habit2000 software [38] was used to time presentation and to record looking times input by the experimenter. A camera, situated through a small circle on the black-card surrounding the screen was used to record looking behaviour. Each session was recorded so the data could be re-coded by a second observer. 

The auditory stimulus that was presented during collision was a natural sound of a billiard ball hitting a wooden cube. Audition C.C. (2016), Adobe Systems was used to amplify the sound. This stimulus was used for all test events for all experiments. The stimulus had a duration of 0.3 s, an acoustic amplitude of 50–58 dB (range) and an auditory frequency of 32–851 Hz (range). The impact sound (i.e., when the ball hit the cube) was 851 Hz and 58 dB.

### 2.3. Procedure

Infants were randomly assigned to one of four groups (N = 4 per group) according to the order in which events were watched:

Group one: B-C-E-D;

Group two: E-D-B-C;

Group three: C-B-D-E;

Group four: D-E-C-B.

Following parental consent to take part in the experiment after being informed about the study, infants were subdivided into four groups (N = 4; M = 2, F = 2, per group) to counterbalance the order of the test events. Infants viewed the computer-generated collision events in the specific order outlined above.

Infants first viewed the habituation trials. The habituation trials were viewed till successful habituation or completion of all nine trials. One habituation trial was presented in a loop for a maximum of 60 s. The duration of the habituation trial was infant dependent. A rattle was presented after the end of each habituation trial to direct infants’ attention back to the screen. Next, infants were presented with the four test trials in that specific order depending on group they were assigned to. Infants saw the test trials in a loop for a maximum of 60 s. The duration of the test trials was again infant dependent. A rattle was presented after the end of each test trial to direct infants’ attention back to the screen.

## 3. Experiment 1 Results

All data were analysed using SPSS version 21 (IBM North America, New York, NY, USA). The looking time data for both habituation and test trials were not normally distributed thus a log transformation was performed. Data were tested for normal distribution by Shapiro–Wilk’s test and for homogeneity of variance using Levene’s test. Following confirmation of parametricity, an analysis of variance (ANOVA, San Francisco, CA, USA) with repeated measures was used to test for differences in log looking time with size (large or small) and congruency (congruent or incongruent) as within-subjects factors. Subsequently, paired samples t-tests with Bonferroni corrections were performed to locate differences. We report alpha levels as exact *p* values, without dichotomous interpretation of ‘significant’ or ‘non-significant’ as advised by the American Statistical Association [39]. Effect sizes are reported using partial eta squared (ηp^2^) for ANOVA and Cohen’s *d* for pairwise comparison. ηp^2^ was interpreted as small (0.02), medium (0.13), and large (0.26) effects. Cohen’s *d* was interpreted as small (0.2), medium (0.5), and large (0.8) effects [40]. Our inferences were guided by a combination of *p* values, mean differences, and effect sizes. Differences were generally considered meaningful if *p* < 0.1, ηp^2^ > 0.13 and Cohen’s *d* > 0.5 [41,42,43]. Figures were generated in GraphPad Prism (GraphPad Prism 8.4.3, GraphPad Software Inc., San Diego, CA, USA) and display grouped dot plots with mean and 95% confidence intervals (CIs) as recommended by Drummond and Vowler [44] and Weissgerber et al. [45]. Data are presented in text as mean ± SD. 

### 3.1. Habituation Trials

Infants’ looking time during the last four habituation trials were analysed with a 4 × 4 mixed-model analysis of variance (ANOVA) with order group (1,2,3, or 4) as a between-subjects factor and habituation trials (1–4) as a within-subject factor. The analysis revealed a main effect of habituation trial number (F(3, 36) = 11.77, *p* < 0.001, np^2^ = 0.48), demonstrating a decrease in looking time across habituation trials. There was an effect of order group (F (3, 12) = 0.34, *p* < 0.027, np^2^ = 0.44) meaning that looking time across order groups differed. The interaction between order group and habituation was F (9, 36) = 1.01 (*p* = 0.07 np^2^ = 0.15). This means that infants in the different order groups did not differ in their looking times across habituation trials. For clarity, infants were considered habituated to an event when their looking time had reduced to a criterion level (for instance, that the mean looking time for last three trials is less than half of the mean looking time for the first three trials. Once the criterion level was reached for habituation then infants started to view the test trials.

### 3.2. Test Trials

The main effect of size from the ANOVA was F (1,15) = 1.440 (*p* = 0.25, np^2^ = 0.13). The main effect of congruency from the ANOVA was F (1,15) = 3.541 (*p* = 0.079, np^2^ = 0.19). The interaction effect between size and congruency from the ANOVA was F (1,15) = 0.411 (*p* = 0.531, np^2^ = 0.03). In terms of pairwise comparisons, the largest differences were between the large ball congruent and large ball incongruent events (*p* = 0.178, *d* = 0.60), small ball congruent and the large ball incongruent events (*p* = 0.272, *d* = 0.55), and small ball incongruent and large ball incongruent events (*p* = 1.000 *d* = 0.32; Figure 3). All other pairwise differences were trivial (*p* = 1.000, *d* < 0.2).

### 3.3. Explanation of Findings

Findings from Experiment 1 are not in agreement with our hypothesis that infants would look longer at incongruent test trials compared to congruent test trials. The findings in this experiment suggest that infants 10–11 months of age were not successful in matching size of the balls with the propelled distance of the cube. We have inferred this as infants did not consistently look longer at trials which violated expectations. By ‘consistently’, we mean that although there was a meaningful effect of congruency from the ANOVA (*p* < 0.1, np^2^ > 0.13), this was driven primarily by the shorter looking time at the large ball incongruent event. It was hypothesised that infants would look *longer* at this event, as it should violate expectation, rather than looking at the event for a *shorter* time. Infants displayed no difference in looking time between any other trials, so possibly this is a statistical artefact, rather than a true phenomenon. We therefore suggests infants in this age group did not use the size of moving object to cue mass in these collision events.

## 4. Experiment 2 Methods

### 4.1. Participants

A total of 31 participants took part in the study, but due to failure to habituate (N = 4) and successful habituation but failure to watch test events (N = 11), the final sample consisted of 16 participants. Infants that failed to habituate did not display a decreased responsiveness to repeated stimuli. Furthermore, infants that successfully habituated but failed to watch the test events displayed a looking time duration that was shorter than the duration of the collision event taking place, this meant the infant did not see the full collision event. Participants were between aged 304 days to 333 days (*M* = 319, *SD* = 11). All participants were recruited from the database at Lancaster University Babylab. Participants were healthy, full-term infants and received a book for their participation alongside being reimbursed for travel costs. Of these 16 infants, 7 were female (age in days: *M* = 312, *SD* = 9) and 9 were male (age in days: *M* = 324, *SD* = 10).

### 4.2. Materials and Apparatus

Animations were the same as in Experiment 1, with some minor changes. The ball was the same size as the mid-size ball in Experiment 1 (*H* = 60 px, *W* = 60 px), and the cube differed in size (small, mid-size and large). During habituation trials the cube was mid-size (*H* = 60 px, *W* = 60 px), but was small (*H* = 40 px, *W* = 40 px) or large (*H* = 90 px, *W* = 90 px) size during test trials (see Figure 4 and Figure 5). 

### 4.3. Procedure

The procedure was identical to Experiment 1. However, the congruent and incongruent trials for the test trials differed from Experiment 1 (see Figure 5). Infants in this experiment were subdivided into the order groups with the following arrangements:

Group one: A-B-C-D;

Group two: B-A-D-C;

Group three: C-D-A-B;

Group four: D-C-B-A.

## 5. Experiment 2 Results

The same statistical analysis was adopted as for Experiment 1. 

### 5.1. Habituation Trials

The looking time data for both habituation and test trials were not normally distributed thus a log transformation was performed. Infants’ looking time during the last four habituation trials were analysed with a 4 × 4 mixed-model analysis of variance (ANOVA) with order group (1,2,3, or 4) as a between-subjects factor and habituation trials (1–4) as a within-subject factor. The analysis revealed a main effect of habituation trial number (F (3, 36) = 18.75, *p* < 0.001, np^2^ = 0.61) demonstrating a difference in looking time across habituation trials, meaning there was a decrease in looking time across habituation trials. There was no effect of order group (F (3, 12) = 2.03, *p* = 0.16, np^2^ = 0.34) meaning that looking time across order groups did not differ. The interaction between order group and habituation was F (9, 36) = 1.20 (*p* = 0.33 np^2^ = 0.23). This means that infants in the different order groups did not differ in their looking times across habituation trials.

### 5.2. Test Trials

The main effect of size from the ANOVA was F (1, 15) = 1.500 (*p* = 0.240, np^2^ = 0.09). The main effect of congruency from the ANOVA was F (1, 15) = 0.631 (*p* = 0.439, np^2^ = 0.04). The interaction effect between size and congruency from the ANOVA was F (1, 15) = 0.117 (*p* = 0.737, np^2^ = 0.01). In terms of pairwise comparisons, the largest differences were between the small cube congruent and the large cube incongruent events (*p* = 1.000, *d* = 0.35), the large cube congruent and small cube congruent events (*p* = 1.000, *d* = 0.31), and small cube congruent and small cube incongruent events (*p* = 1.000 *d* = 0.24; Figure 6). All other pairwise differences were trivial (*p* = 1.000, *d* < 0.2).

### 5.3. Explanation of Findings

Findings from Experiment 2 are not in agreement with our hypothesis that infants would look longer at incongruent test trials compared to congruent test trials. The findings in this experiment suggest that infants 10–11 months of age were not successful in matching size of the cube with the propelled distance of the cube. We have inferred this as infants there was no meaningful effects of congruency or size from the ANOVA (*p* > 0.1, np^2^ < 0.13), and pairwise differences were small at most (*d* < 0.5). We therefore suggest infants in this experiment did not use size of stationary object to cue mass in the collision events.

## 6. Discussion

Findings of experiments in this paper suggest no differences in looking time between congruent and incongruent test trials for 10-to-11-month-old infants, which was contradictory to our hypotheses that infants would display longer looking times at incongruent test trials compared to congruent test trials. In the first experiment, whereby the moving object changed size but the stationary object remained constant, infants did not discriminate between the sizes and their congruent and incongruent distances in line with our hypotheses. Similarly, results from the second experiment in which the moving object remained constant in size and the stationary object changed in size produced analogous outcomes. The first experiment presented herein was a replication of the Sanal-Hayes et al. [35] experiment, but with 10–11 month-olds rather than 6–7 month-olds. The second experiment presents a first attempt to examine infants’ perception of size of stationary objects in collision events using the methodology of Sanal-Hayes et al. [35]. Both experiments suggest infants of 10-to-11 months of age fail to consider size and its mass cues of either the moving, or the stationary object in collision events.

Literature to date suggests infants of 10-to-11 months of age perceive object size and its mass cues in animated collision events [34]. Sanal-Hayes et al. [35] demonstrated that infants of 6-to-7 months of age can differentiate sizes in animated collision events, but cannot perceive the mass cues of size (small or large moving object) in terms of the displaced distance of a stationary object after collision. The experiments by Sanal-Hayes et al. [35] and Hohenberger et al. [34] differ, in that Hohenberger et al. [34] utilised self-propelled animated collision events. This means objects were set into motion without an external force such as a hand. Conversely, we previously incorporated an agent in form of a hand which sets objects in motion [35] and for that reason, test events cannot be described as self-propelled. This is an important methodological consideration as infants fail to perceive causality when events are self-propelled [17].

Kotovsky and Baillargeon [20] argue 6-to-7-month-olds can match size of a moving object with propelled distance of stationary object in real-life objects. Infants of 6-to-7 months of age expected a larger cylinder to displace a toy bug to endpoint of screen, demonstrated by longer looking time at the event which violated expectation; small cylinder performing the same behaviour. Infants match this mentioned event with the large size cylinder as opposed to small size cylinder [20,34]. However, these findings are based on only one distance, the endpoint of the screen (longer distance). Sanal-Hayes et al. [35] examined whether 6-to-7-month-olds would perceive this relationship when both small or large moving object could displace a stationary object to short (before midpoint) or long (endpoint) distances. This task is far more complex than the simple real-life collision events suggested by Kotovsky and Baillargeon [20]. Furthermore, the task of the current experiment was more advanced than the computer-generated collision events suggested by Hohenberger et al. [34]. Based on the findings of Sanal-Hayes et al. [35] concluded that the results with the age group of 6–7 months olds might suggest the task of perceiving mass cues of size of moving object in advanced computer-generated collision events might have been ambiguous or complex for this age group. There are a number of variables that infants need to consider such as (a) object size of the balls and the cube assessed separately, (b) the properties of the balls and the cube assessed in relation to one another, and (c) the likely force one object with a certain property will exert on another object with a certain property. As such, the variables involved in perceiving the object size and their cues to mass in collision events might require advanced reasoning beyond that of this age range. Replication with an older age group suggest that the task might be ambiguous for this age group as well. Future studies might benefit from examining this in older infants or children to determine when this perception manifest. 

Experiment 2 was the first to examine 10- to 11-month-old infants’ perception of size of the stationary object in collision events. The size of the stationary object in collision events have not been studied to our knowledge. Previous studies have studied infants’ expectation of the stationary and moving object [29]. Leslie and Keeble [29] claim that infants around 6 months of age are successful in registering the reversal of the collision events. The reversal of the collision events is when the stationary object propels a moving object to move in the opposite direction [29]. Despite this, no studies to date have examined the size of the stationary object after infants successful understanding of the size of the moving object [20,34]. The second experiment differ from the previously mentioned studies, since it adapted the methodology of Sanal-Hayes et al. [35], thus could have been perceived as advanced beyond this age group. Infants need to consider object size of moving object and cube separately, then in relation to one another and the force each object exerts on the other object depending on certain object properties. Experiment 2 was an easier version of the experiment 1 in this paper. This can be explained by the encoding differences of the events in these experiments. In the second experiment, the cube is present during the entire collision event and can be contrasted with the other cubes below the ramp. Furthermore, the moving object that is off view at certain times has a constant size throughout all events. However, in the first experiment, the moving object is not present at the start of the test event but is later presented by a hand that puts it on the ramp, before it rolls down. This might not give infants enough time to encode and distinguish between the size of the moving object and the two other objects below the ramp. For that reason, the size relationship in the second experiment can be understood easily compared to the first experiment. Furthermore, it is a more complex matter to relate size of one moving object to distance propelled of another object (experiment 1) as opposed to relate size of one stationary object to distance propelled of this same object (experiment 2). Future studies could benefit from extending the study by Kotovsky and Baillargeon [20] but manipulate the stationary object solely. 

In conclusions, results from both experiments presented herein suggest infants fail to perceive mass cues of object size of moving object and object size of stationary object in complex collision events in which objects can be displaced two different distances. This suggest the computer-generated events might have been too ambiguous or complex for this age group. Future studies can benefit from examining these relationships in either real-life objects replicating findings of Kotovsky and Baillargeon [20] or use simplified computer-generated collision events in line with Hohenberger et al. [34] to examine the mass cues of object size of moving and object size of stationary object in collision events. 

## Figures and Tables

**Figure 1 behavsci-13-00056-f001:**
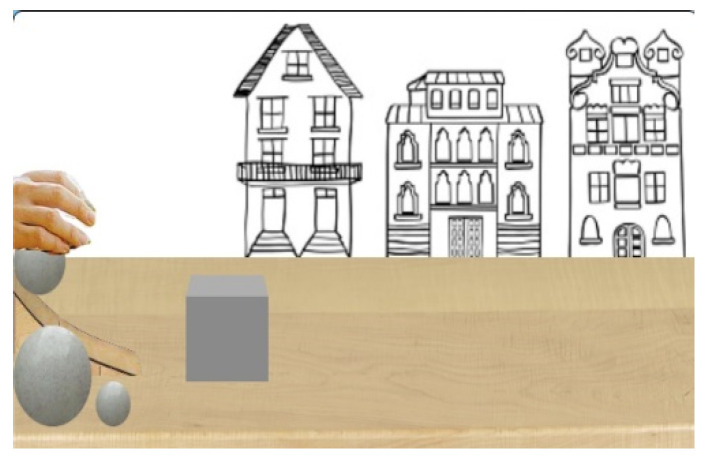
The hand placing the ball on the ramp.

**Figure 2 behavsci-13-00056-f002:**
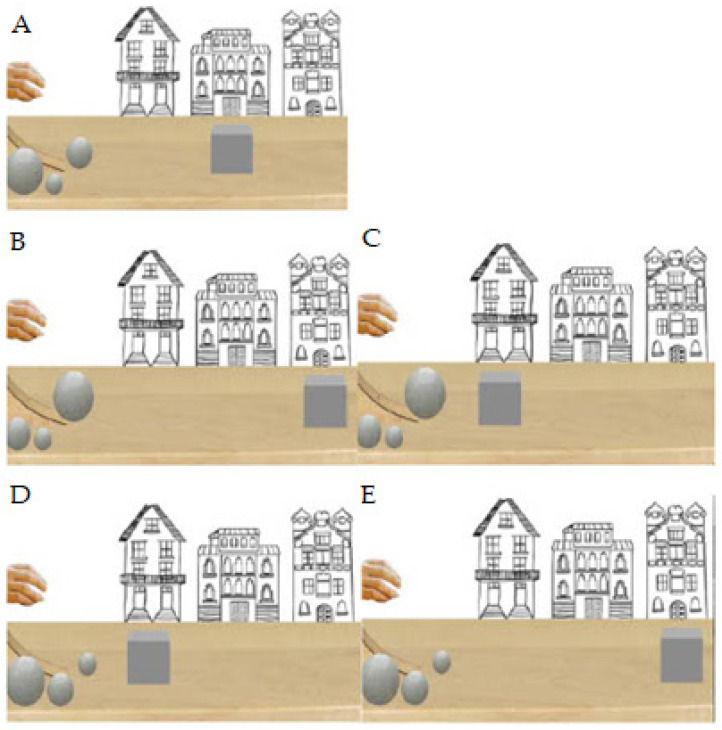
(**A**) Habituation event, (**B**) Large ball congruent, (**C**) Large ball incongruent, Bottom: (**D**) Small ball congruent, (**E**) Small ball incongruent event outcomes.

**Figure 3 behavsci-13-00056-f003:**
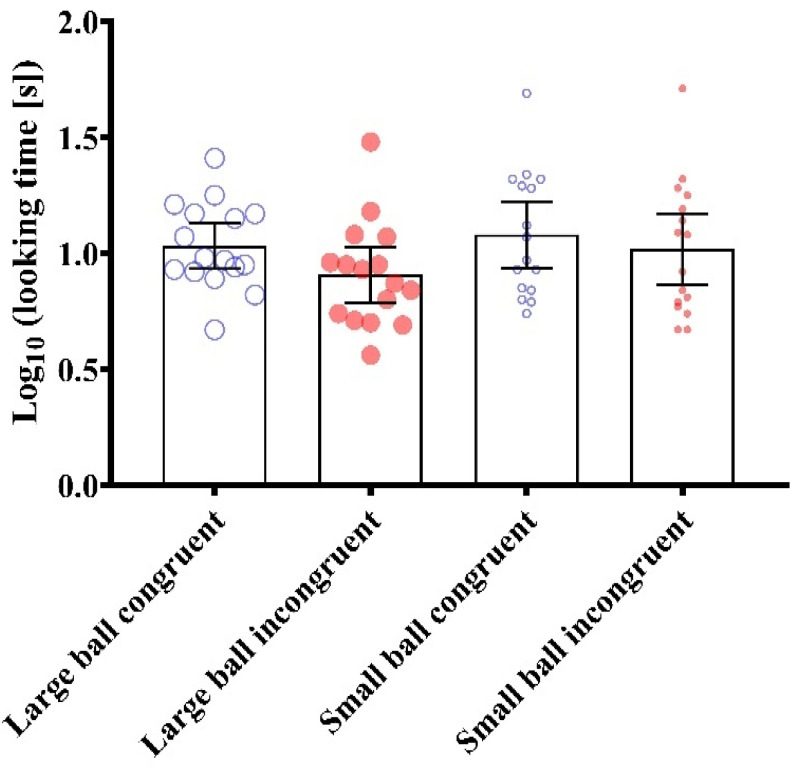
Looking time for small and large ball congruent and incongruent test events. Data are presented as grouped dot plots and mean and 95% confidence intervals. AU = arbitrary units.

**Figure 4 behavsci-13-00056-f004:**
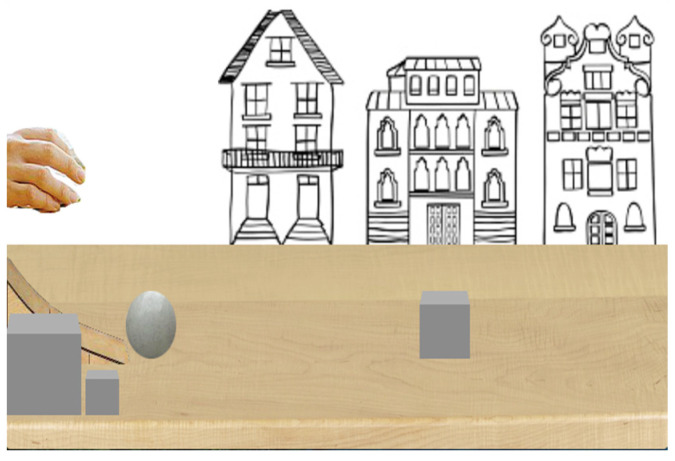
The habituation event infants watched before test events.

**Figure 5 behavsci-13-00056-f005:**
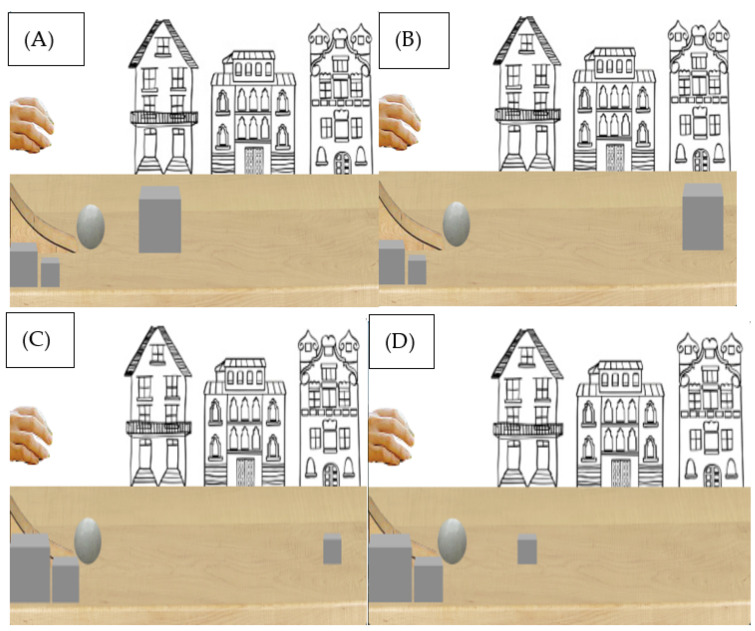
(**A**) Large cube congruent, (**B**) Large cube incongruent, (**C**) Small cube congruent, (**D**) Small cube incongruent test events.

**Figure 6 behavsci-13-00056-f006:**
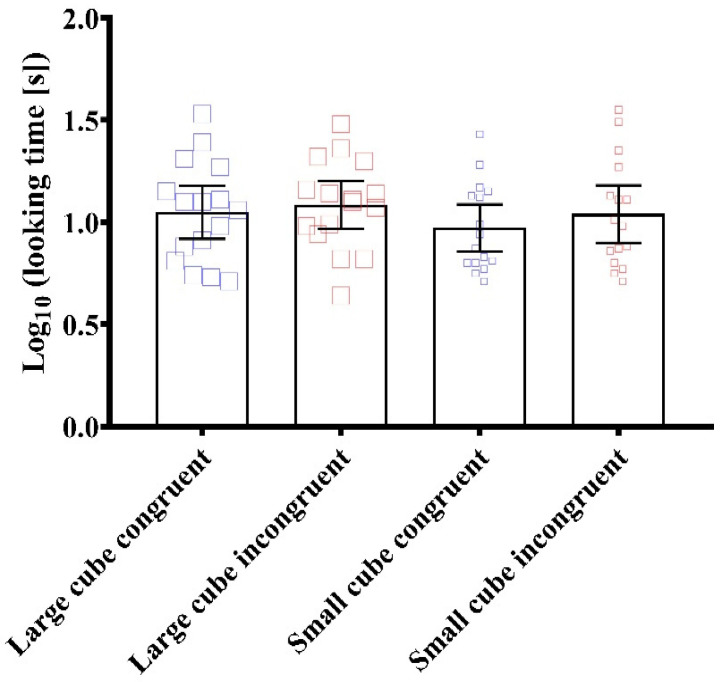
Looking time for small and large cube congruent and incongruent test events. Data are presented as grouped dot plots and mean and 95% confidence intervals. AU = arbitrary units.

## Data Availability

Data will be made freely available on request.

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
