# Peer review of "Perception of Size and Mass Relationships of Moving and Stationary Object in Collision Events in 10-to-11-Month-Old Infants"

_behavsci, 2023, doi:10.3390/bs13010056_

Round 1

Reviewer 1 Report

Dear authors,

This work, on infants' perception of size and mass and their physical effects on propulsion of objects of different sizes, adds to the literature on this topic by (1) replicating an existing paradigm with an older age group, and (2) with the same age group, addressing a reversal of this paradigm in which the size of the displaced, rather than the displacing, object varies. There were no differences in looking time to the test events in either experiment 1 or 2 on the basis of object size, distance travelled, or the interaction between size and distance. The results are interpreted to mean that infants do not perceive the mass cues provided by object size and distance travelled, and it is suggested that the events presented in the stimuli were too complex.

The work is presented succinctly and addresses a clear gap in the literature through the "reversal" of which object's size is manipulated. The authors report their results thoroughly. I have a number of concerns that would need to be addressed before publication, and a couple of minor suggestions.

Major:

1. The Introduction could be improved and grounded more solidly in theory. It reviews the literature on the topic in quite a constrained way, discussing only papers using paradigms that are tightly linked to the experiments at hand. Perhaps not every reader will be immediately interested in the specific question of whether infants perceive the effect of mass of one object on displacement of the other (or the effect of an object's mass on its own displacement) - but they may be interested in the question of how infants perceive and begin to comprehend the physical world? An introductory paragraph telling the reader why empirical questions like the ones investigated here are important or useful to the study of infant development more generally would be good. Likewise, an overview of relevant theory alongside the relevant empirical work would be useful.

2. I would also like to see more information on why the Experiment 2 reversal manipulation was used (other than it not being done before) and hypothesised results of this manipulation (especially as they differ from Experiment 1 hypotheses).

3. I would like to see the Introduction discuss the conflicts in the literature e.g. 5.5-6.5 month-olds could perceive size (mass) effects on an object in one paper (Kotovsky & Baillargeon), but 6-7-month-olds could not in Sanal-Hayes et al. Why one result in one study, but a different one in the next? (e.g. could it have been a size-effect in K&B since they didn't manipulate distance?).

A thorough discussion of these conflicts would be quite useful especially in terms of the authors' final interpretations of their results as the consequence of complex stimuli rather than prior results not replicating with a new sample or in a slightly different paradigm. I note that Kotovsky & Baillargeon also did their paradigm with 10-11-month-olds (1994) and found that infants of this age did perceive size to affect distance a stationary object was displaced after collision. This should probably be cited.

4. The authors say that they do not distinguish between significant and non-significant results on the basis of p-values but all of the results considered to show an effect are p < 0.05, and results above this threshold are considered to show no effect (e.g. p = 0.07 on the interaction between habituation trial number and test order group in Experiment 1). Some p > 0.05 results with small effect sizes (e.g. Exp 1 main effects of size and distance) are interpreted as showing no effect without reference to what a small effect size means in this context. I have no issue with either using or eschewing conventional thresholds as long as results are stated in full (as they are here!) but if the authors state that they're explicitly not using conventional significance thresholds, yet present evaluative statements on the results, the criteria for these evaluations should be stated.

5. The analyses on overall looking time to the habituation trials are good to have, particularly to show that there were no systematic differences in habituation that might have a knock-on effect on results across the test trial orders. However it is puzzling that the effect of group on test trial looking time is only a main effect. With four trials, including incongruent ones, and no re-habituation in between, is it not possible that infants would adjust their assumptions about the mass of the balls? I am reminded of work by Gottwald & Gredeback that suggests infants of slightly older age can make inferences about object weight that go beyond visual cues.

I don't know if there are sufficient participants to provide the power needed to look at the interaction between test trial number, size, and distance, but if not perhaps the authors could (1) collapse the conditions into congruent versus incongruent, OR (2) trial orders into congruent-first and incongruent-first, OR (3) look at first trials only OR (4) plot out descriptives by trial number and condition, to demonstrate that the assumptions of the habituation hadn't been erased by the time of the final trial.

6. Adult participants are mentioned in a few places but there's no information on sample size or adult results. Are the adult results published elsewhere? If so a reference should be given; otherwise the results ought to be reported here.

Minor:

1. There are some issues around subject-verb agreement that are very minor and might be picked up by another round of proofreading. However the abstract is quite difficult to read for this same reason (e.g. "the paper" mentioned in one sentence, but "their results" rather than "its results" in the next) and should be revised.

2. Figure 1 isn't referenced in the text as far as I can see.

3. It might be nice to add another panel to Figure 1, showing the hand placing the ball on the ramp.

4. You could note that the balls differed in (inferred) volume from one another with a particular constancy.

5. A note on the choice of age group would be good. Did you select this group just because the 6-7-month-olds couldn't distinguish the size/distance relationship and were looking for an older age group, or was there another reason based on the literature?

Reviewer 2 Report

First and foremost, may I commend the authors for their efforts! The manuscript looks more like a report of disciplinary physics experiments though! Perhaps, future developments in this focused area of research would be useful for both the social sciences (policy and development programs for infants) and medical physics.

Furthermore, in my own opinion the manuscript is about the perception of the sampled participants, and the rest parts are concerned largely with physical objects in the way of a physics class, though with an intermediary illustrating software (/animation). In other words, the research is about infants’ perception of the size and mass relationship of moving and/or stationary, and collision objects. And generally, the research seems very scientific and well reported.

Title: The giving title seems to have completely captured the essence of the manuscript.

Abstract:  Very good abstract but long!

Keywords: Can the Keywords be arranged in an alphabetic order? What do the authors mean by “infant physics”? I think you need to tidy up what you include as keywords. There are fine examples in very good publications

Introduction: The introduction section seems very okay. However, I will recommend language editing for the whole manuscript. For example, the sixth sentence in the first paragraph looks disconnected, and the first sentence in the second paragraph (particularly, “size cue mass” cues mass or mass size) are not very clear and easy to understand.

Methods: The method for achieving the aim of the research is seemingly very adequate, scientific and well executed.

Result: The result too is seemingly well presented scientifically.

Discussion: The discussion section (and conclusion) focused well on the result and the aim of the research. I think there is a good marriage of the title with the result and discussion section.

References: Lastly, I think the references are so few and typical of the physical sciences! But the editor is in a good position to decide for the journal’s standard. (A journal for the social sciences!)

Round 2

Reviewer 1 Report

Dear authors,

Thank you for engaging with the original review comments, they have been addressed satisfactorily for the most part. I have a few extant comments that are minor in nature:

1. The logic of some of the new insertions is not explained adequately. For example on page 3, the sentence "This relationship in collision events is also easier to perceive due to encoding differences" is presented without any explanation or justification. What are the encoding differences that make it easier to perceive, and why do they do so? The next sentence (starting "Furthermore,") does a good job of explaining why keeping the moving object's size constant makes the infants' task simpler - do the same for the one about "encoding differences".

2. Give me more detail in the response to the original comment 3. The differences between the studies might be explained by the fact that stimuli were computer generated in one, and live in the other. Why should that make a difference? (I have my guesses - but you should tell the reader what you're thinking). Likewise, the sentence "
Furthermore, the complexity of the collision events also differed, one examined two sizes on one travelled distance of stationary object, whereas another examined two sizes on two travelled distances of stationary object" on page 3 makes sense - now just add another sentence explaining why this would make a difference (e.g. more likely to see a difference in looking time when infants had less info to process?)

3. "Main effect of habituation trial number" might be easier to understand than "main effect of habituation" (page 7)

4. The 1994 Kotovsky and Baillargeon paper isn't cited in the new addition based on original comment 3 - it's still the 1998 one.

5. The list of references hasn't been updated to reflect the new insertions.
